# Preserving friendships in school contacts: An algorithm to construct synthetic temporal networks for epidemic modelling

**Lucille Calmon**[1], **Elisabetta Colosi**[1¤], **Giulia Bassignana**[1], **Alain Barrat**[2‡], **Vittoria Colizza**[1,3‡]*

**1** Sorbonne Université, INSERM, Pierre-Louis Institute of Epidemiology and Public Health (IPLESP), Paris, France, **2** Aix Marseille Univ, Université de Toulon, CNRS, CPT, Turing Center for Living Systems, Marseille, France, **3** Department of Biology, Georgetown University, Washington, District of Columbia, United States of America

¤ Current address: Bocconi University, Dondena Centre for Research on Social Dynamics and Public Policy, Milan, Italy
‡ These authors contributed equally to this work.
* vittoria.colizza@inserm.fr

**Data Availability Statement:** Empirical contact networks used as base for the synthetic contacts are publicly available as part of (Mastrandrea at al, PLoS ONE, 2015) on the web-page http://www.

## Abstract

High-resolution temporal data on contacts between hosts provide crucial information on the mixing patterns underlying infectious disease transmission. Publicly available data sets of contact data are however typically recorded over short time windows with respect to the duration of an epidemic. To inform models of disease transmission, data are thus often repeated several times, yielding synthetic data covering long enough timescales. Looping over short term data to approximate contact patterns on longer timescales can lead to unrealistic transmission chains because of the deterministic repetition of all contacts, without any renewal of the contact partners of each individual between successive periods. Real contacts indeed include a combination of regularly repeated contacts (e.g., due to friendship relations) and of more casual ones. In this paper, we propose an algorithm to longitudinally extend contact data recorded in a school setting, taking into account this dual aspect of contacts and in particular the presence of repeated contacts due to friendships. To illustrate the interest of such an algorithm, we then simulate the spread of SARS-CoV-2 on our synthetic contacts using an agent-based model specific to the school setting. We compare the results with simulations performed on synthetic data extended with simpler algorithms to determine the impact of preserving friendships in the data extension method. Notably, the preservation of friendships does not strongly affect transmission routes between classes in the school but leads to different infection pathways between individual students. Our results moreover indicate that gathering contact data during two days in a population is sufficient to generate realistic synthetic contact sequences between individuals in that population on longer timescales. The proposed tool will allow modellers to leverage existing contact data, and contributes to the design of optimal future field data collection.

sociopatterns.org/datasets/high-school-contact-and-friendship-networks/. Python code generating synthetic contacts with the friendship-based and class-mixing-based approaches can be found in the github repository https://github.com/EPIcx-lab/synthetic-school-contact-networks.

**Funding:** This study was partially funded by: EU Horizon 2020 grant MOOD (H2020-874850, paper 082) to LC, EC, VC; Horizon Europe grant ESCAPE (101095619) to VC; Horizon Europe grant VERDI (101045989) to EC, VC; the Agence Nationale de la Recherche (ANR) project DATAREDUX (ANR-19-CE46-0008) to AB, VC. The funders had no role in study design, data collection and analysis, decision to publish, or preparation of the manuscript.

**Competing interests:** The authors have declared that no competing interests exist.

## Author summary

Face-to-face contacts occur between individuals throughout day-to-day activities. These contacts form a network of opportunities for the spread of diseases, such as COVID-19 or influenza. Modelling the transmission of respiratory viruses along such networks can provide insights on the efficacy of mitigation measures. To be actionable, such models must be fed with realistic contact data covering the time scale of an epidemic. However, high-resolution data on temporal contacts are most often limited to recording windows of a few days. In this manuscript, we propose two methods (the friendship-based and class-mixing-based approaches) to synthetically extend high-resolution contact data recorded in a secondary school. The class-mixing-based approach takes into account the organisation of students in classes, the overall activity patterns dictated by the school timetable, and the observed contact duration distributions. The friendship-based approach is further optimised to preserve friendships among the students, which induce repeated contacts with correlated durations. Using an agent-based model describing the spread of COVID-19 in a school, we show that taking into account friendships when creating synthetic contact data leads to transmission chains among students that differ substantially from those obtained with class-mixing-based contacts. Both types of contacts instead predict similar transmission pathways between classes. Our manuscript thus provides tools to generate improved model inputs from existing data with limited temporal scales, and an evaluation of the impact of preserving friendships on simulated COVID-19 outbreaks.

## Introduction

Face-to-face contacts between individuals represent crucial transmission paths for respiratory viruses, such as SARS-CoV-2, influenza or RSV (respiratory syncytial virus) [1]. These contacts occur throughout day-to-day life, e.g., in public transport, among children in schools, coworkers in workplaces, household members, as well as in the wider community, creating opportunities for the spread of diseases. The global contact network among individuals in a population, which thus approximates possibilities for transmission, remains challenging to measure [2–4]. Nonetheless, an increasing number of research groups have developed ways to automatically record high-resolution empirical contact networks in a variety of settings, particularly in various types of schools [5–10], but also in a university [11], in an office building [12], in a conference [13], in hospital wards [14–17], in households in rural villages [18] or on a cruise-ship [19]. These empirical contact networks in turn can provide valuable input to inform models of infectious disease spread and to design or evaluate intervention strategies [5, 15, 17, 20–25].

Several characteristics of the network of contacts impact the patterns of a spread unfolding on that network. For instance, clustering (the tendency of the contacts of a person to be in contacts themselves) [26, 27], heterogeneity in the number of contacts per individual [28], contact duration [9, 29], and the repetitions of contacts in successive periods (e.g., days) [13, 30–33] all influence the disease spread. The role of contact repetition [26, 27] in particular has been recently considered [33, 34]. When generating contact networks from contact matrices [35, 36] at a daily temporal scale to inform disease spreading models, not taking into account the fact that a non-negligible fraction of contacts are repeated from one day to the next (up to 35% of reported contacts according to some surveys [31]) can lead to overestimated attack rates [33]. Additionally, contact repetition (over different days) and retention (the tendency of people to remain in contact over time with an individual) can also affect the identification of

superspreaders and superspreading events [34]. When building synthetic data on contacts between individuals to feed e.g. agent-based models of disease spread [13, 37, 38], correctly taking into account the presence of repeated contacts is therefore crucial.

Despite the increased availability of high-resolution data on contact patterns [3, 5–14, 16, 19], each data set remains limited to a given specific context and data collection period, and most often to a relatively short data collection time window of a few days. On the other hand, agent-based models need to be simulated on long time scales and their results should not depend on the specificity of a data set collection time [25, 29]. These models thus need to be fed by realistic synthetic data covering arbitrarily long time scales. Such inputs are obtained by longitudinally extending recorded data, often by simply repeating the empirical data over and over [13, 21, 22, 24, 39], which means that each and every contact or contact pattern is repeated periodically in such synthetic data, with no variation. So far, few works have explored the potential of leveraging data on the actual amount of repetition of contacts between different periods (e.g., days) to extend longitudinally existing data sets in a way that respects the balance between repeated contacts and more casual, randomly occurring ones. This is particularly relevant for contacts recorded in educational settings where (1) various respiratory viral infections circulate among students [40] that densely mix in an indoor context [6, 7] and (2) strong social ties due for instance to friendships and associated with longer contacts [41] drive behaviour, inducing repeated encounters with correlated characteristics that co-exist with casual interactions [7, 8].

Here, we tackle this issue and present and illustrate methods to use empirical contact data recorded during a few days to generate long-term synthetic contact data with realistic statistical properties and on arbitrarily long timescales, in the context of a secondary school [8]. We specifically design and compare two mechanisms for this purpose: In the "Friendship-based approach", we generate synthetic contacts that take into account data on the repetition of contacts across different days, which we interpret as a sign of probable friendship between the corresponding individuals; in the "Class-mixing-based approach" on the other hand, we preserve the mixing patterns between classes in each day of synthetic data, but do not consider any other memory effects between days (i.e., we do not take into account the data on contact repetition). As a baseline, we also consider a simple procedure of looping the empirical contact data.

We then use the three types of obtained synthetic data sets to separately inform a realistic agent-based model of the spread of SARS-CoV-2 in a school [24]. In order to evaluate the impact of taking into account friendship relations in this context, we compare the outcomes of the numerical simulations fed by the various types of synthetic contact data, focusing on the one hand on the distributions of outbreak sizes, and on the other hand on the infection networks and infection trees [42, 43], which summarize the preferential paths along which the disease progresses through the population.

## Methods

### Empirical contact networks

We consider empirical contact data collected among students in the second grade of "Classes Préparatoires" [8]. These classes are offered by some high-schools in France during two additional years to prepare students for entrance examinations to higher-education establishments in France. In the "Lycée Thiers" (Marseille, France) where the data was gathered, the second grade of these classes is organised in 9 classes of 3 specialties (mathematics: 3 "MP" classes, biology: 3 "BIO" classes, and physics-chemistry: 3 "PC" classes). Close face-to-face proximity events were recorded between students in these 9 classes during school hours [7], as described

in [8, 44]: such contacts were recorded between students wearing RFID (Radio Frequency Identification) sensors that exchanged low-power data packets when in close proximity (up to 1–1.5 meters). Data was collected over four and a half days (Monday the $2^{nd}$ to Friday the $6^{th}$ of December 2013), with a high participation rate: out of the 379 students in the 9 second grade classes, 327 participated to the study (86.3% participation rate). The resulting data sets are temporal networks were nodes represent students and temporal edges represent recorded proximity events with a temporal resolution of 20 seconds [44, 45]. Such data can also be expressed as lists of timestamps $t_{ij}$ for each pair $(i, j)$ of individuals who have been in contact. We moreover build, from the data and without loss of information, daily contact networks as follows: for each day $d$ of the deployment, such a network $G_d$ includes students as nodes and encodes the occurrence of contacts between them during that day as contact links, denoted $(i, j)$, that carry both a weight $w_{ij}^d$ and a contact timeline $tl_{ij}^d$. The weight $w_{ij}^d$ is given by the cumulative time recorded in contact between students $i$ and $j$ during day $d$, and the "activity timeline" $tl_{ij}^d$ is given by the list $\{t_{ij}\}_d$ of all contact timestamps between nodes $i$ and $j$ during day $d$ (note that $w_{ij}^d$ is thus simply proportional to the length of the list $\{t_{ij}\}_d$).

Using the daily contact networks, we identify the pairs of students who have been in contact in at least two different days of the data collection. Interestingly, the data also includes information on friendship relations between students, as gathered from a survey [8]. We use the combination of contact data and friendship data to show in the S1 Text that the repetition of contacts across distinct days is an indication of the likely existence of a friendship relationship between the students. We will thus refer in the following to the pairs of students that are in contact during at least two days of the deployment as "friendship links". Note that the links of each daily network $G_d$ will thus be divided into (1) a set $F_d$ of friendship links $(i, j)$ that are observed in at least one other daily network $G_{d'}$ (with $d' \neq d$) and (2) a set $C_d$ of "casual" links between individuals who have been observed in contact only in day $d$.

## Synthetic extension of the contact data

We propose here two methods to leverage the empirical daily contact networks and create synthetic ones among the same students, with high-resolution timelines and preserving several important statistical properties observed in the empirical data. For each day, each method takes as input a daily contact network and can output an arbitrary number of similar, statistically plausible, synthetic daily contact networks. Both methods preserve in particular the class-mixing matrix of the day considered, defined as the number of contact links $N_{AB}$ between each pair of classes $A$, $B$ and within each class $A = B$ during that day. In the first method, which we call the "friendship-based approach", friendship links of the day considered are preferentially retained in the synthetic version of that day. This approach hence preserves the local structure of the students friendships, while additional random contact links mimic stochasticity in contact behaviour (the "casual" links in the data). In a second approach, the "class-mixing-based approach", the distinction between friendship and casual links is not taken into account to create the synthetic data, which only preserve the class-mixing matrix and the overall timeline of when interactions occur within a class or between two classes (to respect e.g. the fact that interactions between classes may take place only when the school schedule allows it).

**Friendship-based approach.** In this approach, we take into account that, as discussed above, the contact patterns among students are not fully random but bear similarities over different days. Contacts between a given pair of students may reoccur in different days [6, 34, 46], in particular between friends, and the contacts total duration (link weights) of friendship links also tend to be larger [7].

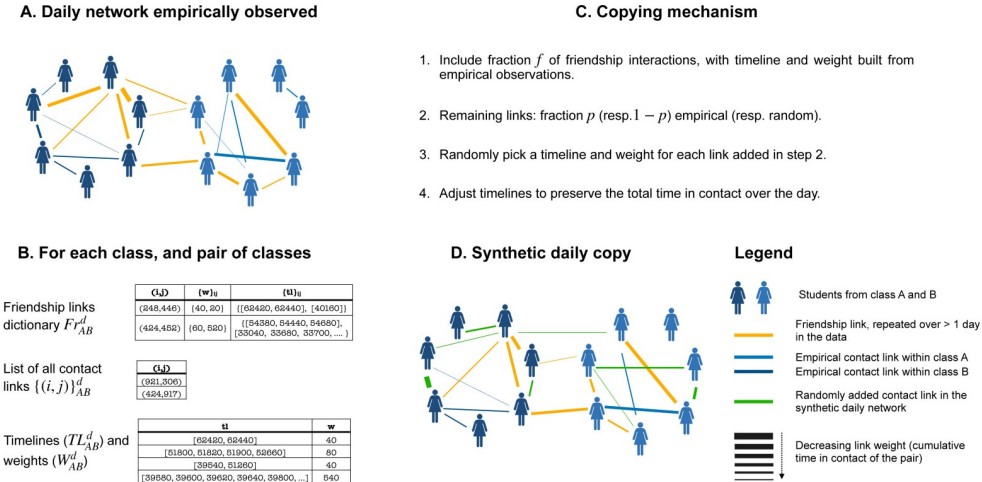

**Fig 1. Schematic representation of the friendship-based approach to synthetic contacts generation.** (A) Daily high-resolution empirical contact networks are schematically represented. They form the starting point (input) of the method. Friendship links are highlighted in orange, while blue links correspond to empirical links occurring only on a single day. (B) Different lists required for the algorithm are shown with examples of entries. Note that the timelines (lists of timestamps) are expressed in seconds from midnight on the initial day of the deployment, and the weights ("w") are expressed in seconds. These lists are specific for each class and pair of classes, and each base day. (C) Description of the different steps of the algorithm. When assigning randomly a timeline and weight to contact links (step 3 of (C)), an entry of the third table is drawn. The algorithm operates class by class, and pair of classes by pair of classes, to generate synthetic contact networks. (D) Schematic representation of the generated synthetic contact networks. These networks inherit properties from the empirical contact networks, such as the number of links per class and between each pair of classes, a fraction of the friendship links (depicted in orange) and of the non-repeated links (depicted in blue). These links are complemented by random links (depicted in green) that were not necessarily observed in the base day.

We thus propose the algorithm below (summarized in Fig 1) to generate synthetic daily contact networks. The algorithm takes as input (1) one day of data collection, that we call the base day and (2) the separation between friendship links and casual links of that day. It can thus be ran independently for each day of data collection once the friendship links have been extracted from the comparison of the various empirical daily contact networks. The algorithm generates contact links class by class, and pair of classes by pair of classes until all classes (and pairs of classes) have been considered. Combining all the contact events generated provides synthetic contacts inheriting the properties of the daily contact network of the base day considered.

For a given base day $d$, the algorithm proceeds as follows to build synthetic contact networks between the students of each pair of classes $A$ and $B$ (and among the students of each class $A = B$).

**1. Inputs from data:** the algorithm uses as inputs the number of contact links $N_{AB}^d$ between classes $A$ and $B$ on the base day, the lists of contact links $\{(i,j)\}_{AB}^d$ between them (i.e., all contact links such that $i$ is in class $A$ and $j$ is in class $B$ or vice-versa), of their weights $W_{AB}^d = \{w_{ij}\}$ (cumulative contact durations) and associated timelines $TL_{AB}^d = \{tl_{ij}\}$. The inputs include as well the friendship links on the base day between individuals of classes $A$ and $B$, which are put in a dictionary of the form $\mathbf{Fr}_{AB}^d[(i,j)] = \{weights : \{w\}_{ij}, \ timelines : \{tl\}_{ij}\}$, where $\{w\}_{ij}$ is the list of weights of $(i,j)$, and $\{tl\}_{ij}$ the list of timelines of $(i,j)$ on all days in which the friendship link is observed in the data.

2. **Taking friendships into account:** Each of the friendship links in $\mathbf{Fr}_{AB}^d$ is included as a contact link in the synthetic network with probability $f$. For each such link $(i, j)$ added in this step, a synthetic weight and timeline are generated from $\{w\}_{ij}$ and $\{tl\}_{ij}$ (see S1 Text for details).

3. **Remaining links:** Additional links are added one at a time. Each such link is either extracted from the list of empirical contact links $\{(i, j)\}_{AB}^d$, with probability $p$, or created randomly (with probability $1 - p$) by choosing at random two students respectively in classes $A$ and $B$. In both cases, with probability $p_{tr}$ we add an extra step to preserve transitivity in the network [47] (the fraction of closed triangles, i.e. of structures $\{(i, j), (j, k), (k, i)\}$ among all possible connected triads, i.e., such that at least $\{(i, j), (j, k)\}$ exist). Specifically, if the transitivity of the current synthetic network between classes $A$ and $B$ is lower than the empirical one, the additional link is chosen (either in $\{(i, j)\}_{AB}^d$ or randomly) such that it closes a triangle; conversely, if the current transitivity is too high, the additional link is chosen in order to create an open triangle. With probability $1 - p_{tr}$, the choice of the additional link is made independently from the transitivity of the network. This procedure of adding links is iterated until the synthetic contact network between classes $A$ and $B$ (or within class $A = B$) contains $N_{AB}^d$ contact links. Each added contact link is then associated a timeline and matching weight selected randomly from $TL_{AB}^d$ and $W_{AB}^d$.

4. **Weight correction:** Once the synthetic contact network between classes $A$ and $B$ (or within class $A = B$) includes the correct number of contact links, timestamps can be added or removed from activity timelines in the network (longer timelines are preferably modified) to ensure that the sum of all link weights in the synthetic contact network between classes $A$ and $B$ (or within class $A = B$) is within a 10% tolerance of the corresponding quantity in the empirical contact network of the base day—see S1 Text for details.

The above steps are repeated for every pair of classes $A$ and $B$ and for each class ($A = B$). The resulting synthetic contact networks are then merged into daily synthetic contact networks involving all individuals present in the base day. By construction, these synthetic contact networks preserve the number of contact links and the total time in contact within and between each class (steps 3 and 4). In addition, as the timelines of the contact links are taken from the lists of empirical ones, the timetable of the base day is also preserved.

In the algorithm, the parameter $f$ controls the average fraction of friendship links that are preserved in the synthetic data. The parameter $1 - p$ controls the amount of stochasticity, by allowing links in the synthetic network between students who have actually not been observed in contact during the base day. In the following, we describe how we tune these parameters ($f$ and $p$), as well $p_{tr}$, to reproduce some features of the empirical networks.

**Parameters optimisation.**   We tune the parameters of the model in order to take into account the day-to-day similarity between daily contact networks observed empirically. To quantify this similarity, we consider the local cosine similarity [48], which measures the similarity in the contact links of an individual $i$ in two daily contact networks (on days $d_1$, $d_2$). Mathematically, it is given by

$$LCS(i, d_1, d_2) = \frac{\sum_{j \in \mathcal{N}_1 \cap \mathcal{N}_2} w_{ij}(d_1) w_{ij}(d_2)}{\sqrt{\sum_{j \in \mathcal{N}_1^*} w_{ij}(d_1)^2} \sqrt{\sum_{j \in \mathcal{N}_2^*} w_{ij}(d_2)^2}} \tag{1}$$

where $\mathcal{N}_1$ (resp. $\mathcal{N}_2$) refers to the set of nodes $j$ in contact with node $i$ on day $d_1$ (resp. $d_2$) with contact link weights $w_{ij}(d_1)$ (resp. $w_{ij}(d_2)$). The sums in the denominators run over $\mathcal{N}_1^*$ (resp. $\mathcal{N}_2^*$) which is the set of nodes in $\mathcal{N}_1$ (resp. $\mathcal{N}_2$) that are also present in the daily network of day

$d_2$ (resp. $d_1$). This local-node-based similarity measure ranges from 0 to 1. When $i$ has contact links with disjoint sets of individuals on days $d_1$ and $d_2$, $LCS(i, d_1, d_2) = 0$. Partially overlapping contact links with different cumulative duration (encoded in the weights $w_{ij}(d_1)$ and $w_{ij}(d_2)$) instead lead to $0 < LCS(i, d_1, d_2) < 1$, reaching 1 only for an individual $i$ with exclusively the same contact links and proportional weights in both days.

The distribution of values of the local similarity between any pairs of data collection days for the participating students [8] exhibits relative maxima at 0 and 1 values but is spread over all possible similarity values, showing a strong heterogeneity of similarity patterns (see Results Section) [30]. We thus tune the algorithm parameters $f$, $p$ and $p_{tr}$ to reproduce this specific shape. Specifically, we perform a grid-exploration for these three parameters to minimise the Jensen-Shannon distance [49, 50] between the synthetic and empirical distributions of local cosine similarities. Details on the optimisation process, the definition of the Jensen-Shannon distance as well as a sensitivity analysis on the distance measure used can be found in the S1 Text. In the following, contact networks generated with this method, using optimised parameters, are called friendship-based (daily) contact networks. These synthetic daily contact networks can also be expressed as lists of temporal contacts between individuals (with the same temporal resolution as the empirical data), which we call friendship-based (synthetic) contacts.

**Class-mixing-based approach.** In the "class-mixing-based approach", the synthetic contacts are generated in order to reflect the empirical mixing patterns between classes observed during the base day considered. However, the algorithm does not take into account the distinction between friendship and casual links, nor the transitivity (see S1 Text for details on the procedure). It is thus approximately recovered instead by the previous algorithm with parameters $f = 0$, $p = 0$ and $p_{tr} = 0$. In particular, the number of contact links within each class and among each pair of classes is preserved, as in the friendship-based approach. However, the contact links between students are renewed each day independently of any previous occurrence. Overall, the synthetic contacts obtained in this approach preserve the class structure of the school, as well as the timetable features, but not the individual friendships.

**Creating synthetic contact data of arbitrary length.** The algorithms described above create synthetic daily contact networks from empirical base days. As the school timetable has typically a weekly periodicity, a natural procedure to create synthetic data covering e.g. $n$ weeks, starting from empirical data collected during one week, is to create $n$ instances of each empirical base day, and to create synthetic weeks using a synthetic instance of each weekday (note that the students are considered isolated during the weekends [21, 24]). In the present case, as, on the Monday, the data was collected only during half a day (Supplementary figures of [8]), we use the Tuesday data as base day to create the Mondays synthetic contact networks. The synthetic contact sequences built using the friendship-based approach applied in this way to the four base days of data (Tuesday, Wednesday, Thursday, Friday) are called "Friendship 4d" in the following, while synthetic contact sequences obtained with the class-mixing-based approach are called "Class Mixing 4d".

As several data collection efforts have been carried out over even shorter timescales [3, 44], we also mimic here such cases, by artificially restricting the number of base days available. For the friendship-based algorithm, the minimal number of base days (in order to separate friendship from casual links) is two, while for the class-mixing-based approach we can even restrict the data to one day. We then create synthetic weekly sequences by combining the synthetic instances obtained from the various base days as detailed in Table 1.

In addition, we considered as a baseline synthetic contact sequences built exclusively by repeating empirical ones, i.e., by looping over empirical contact data as done in many previous works. We call the resulting data looped contacts, where the repeated empirical data can include from one to four of the available daily contact networks. Contact link weights of all

**Table 1. Contact sequences used in the analysis.**

| Sequence name | Approach | Base days | Weekly sequence | Total number of days |
|---|---|---|---|---|
| Friendship 4d | Friendship-based | 2,3,4,5 | [2, 2, 3, 4, 5] | 120 days |
| Friendship 3d | Friendship-based | 2,3,4 | [2, 2, 3, 4, 3] | 120 days |
| Friendship 2d | Friendship-based | 2,3 | [2, 2, 3, 2, 3] | 120 days |
| Class Mixing 4d | Class-mixing | 2,3,4,5 | [2, 2, 3, 4, 5] | 120 days |
| Class Mixing 3d | Class-mixing | 2,3,4 | [2, 2, 3, 4, 3] | 120 days |
| Class Mixing 2d | Class-mixing | 2,3 | [2, 2, 3, 2, 3] | 120 days |
| Class Mixing 1d | Class-mixing | 2 | [2, 2, 2, 2, 2] | 120 days |
| Looped 4d | Looping data | 2,3,4,5 | [2, 2, 3, 4, 5] | 120 days |
| Looped 3d | Looping data | 2,3,4 | [2, 2, 3, 4, 3] | 120 days |
| Looped 2d | Looping data | 2,3 | [2, 2, 3, 2, 3] | 120 days |
| Looped 1d | Looping data | 2 | [2, 2, 2, 2, 2] | 120 days |

The base days and weekly sequence refer both to the underlying four days in the data set as follows: Tuesday (2), Wednesday (3), Thursday (4) and Friday (5). Only four full days are available, so that we systematically build the synthetic contacts on Mondays using the Tuesday base day, in order to maintain a weekly periodicity. For all cases we create synthetic contact sequences of total length 120 days. These contact sequences are composed of weekly sequences of each five days of synthetic contacts and two weekend days where the students are treated as isolated.

sequences are uniformly rescaled to match the total daily interaction time in the "Looped 4d" contacts. This ensures the comparability of epidemiological outputs on contact sequences generated from different sequences of base days.

We denote in the following the various synthetic contact sequences by $ct_x$ (where $x$ is one of the sequence names of Table 1).

## Transmission model

In the following, we leverage on the contact sequences from Table 1 to feed the stochastic agent-based model from Ref. [24] and simulate numerically the spread of SARS-CoV-2 in the school population. In this model, when a transmission event occurs, the exposed individual (E) becomes infectious and pre-symptomatic ($I_p$) after a time $\tau_E$. The pre-symptomatic phase lasts $\tau_p$, after which the individual enters either a sub-clinical ($I_{sc}$) or clinical ($I_c$) infectious state, which lasts $\tau_I$ before recovery (R). The durations of each stage ($\tau_E$, $\tau_p$, $\tau_I$) are drawn at random for each individual from Gamma distributions parameterised following the literature [24]. The probability of developing a clinical (versus a sub-clinical infection) is instead a fixed parameter.

The per unit-time transmission probability when a susceptible individual is in contact with an infectious one depends on the transmission rate $\beta$ and on the relative infectiousness and susceptibility of the individuals in contact, determined from infectious status. We model a vaccinated student population (homogeneous 50% coverage) by reducing the relative infectiousness of vaccinated individuals by 20% and their relative susceptibility to infection by 50%. Moreover, we model partial immunity from previous infection by SARS-CoV-2 pre-Omicron variants in 40% of the population. The susceptibility of immune individuals to infection is reduced by 81%. As time progresses, contacts encoded in the contact sequence considered are replayed in the numerical simulation. To increase computational efficiency, high-resolution 20 seconds contacts are aggregated in 15 minutes steps: each resulting contact is weighted by the fraction of time actually spent in contact over the 15 minutes for each pair. Each such contact event in which an infectious individual is interacting with a susceptible one represents an opportunity of transmission for the simulated disease.

We consider a simple spreading scenario in the absence of control measures initiated by a single infectious individual (the seed) among the students. The transmission rate $\beta$ is tuned as in [24], to achieve an effective reproductive number in the population with partial vaccination and immunity $R = 1.5$ (see S1 Text for details on the calibration).

### Extraction and comparison of infection pathways

For each contact sequence ($ct_x$, in Table 1) and for each initial seed $s$, we build an **infection network** [42, 43], denoted $\mathcal{G}_{inf}(s, ct_x)$, by aggregating transmission chains obtained from different simulations (Fig 2A and 2B). These infection networks are directed, in order to encode transmission events from $i$ to $j$ and from $j$ to $i$ separately. Both occurrences indeed correspond to different transmission chains. Each may occur with a different probability (depending on vaccination statuses of the pair, disease stages during which the contact occurs). Each directed edge $\ell = (i, j) \neq (j, i)$ from student $i$ to $j$ is weighted by the occurrence probability of a transmission event from $i$ to $j$ (denoted $p_\ell(s, ct_x)$), estimated by the fraction of simulations in which such an event is observed [43]. This probability depends not only on the per unit time probability of transmission (see S1 Text), but also on the relative positions of $i$ and $j$ in the contact network. The latter impacts in particular whether $i$ is infected before or after $j$.

Infection networks form an aggregated representation of the outbreak, in which a large fraction (or even all) of the contact links between individuals in the population are represented. We thus also consider summarized versions, the **infection trees** (Fig 2C). Each infection tree, denoted $\mathcal{T}_{inf}(s, ct_x)$, is given by the (directed) maximum spanning tree of the corresponding infection network, as described in [42]. It approximates the "most likely" infection pathway from the seed $s$ to each node, by retaining edges $\ell$ that form a directed tree, such that their summed $p_\ell(s, ct_x)$ is maximum.

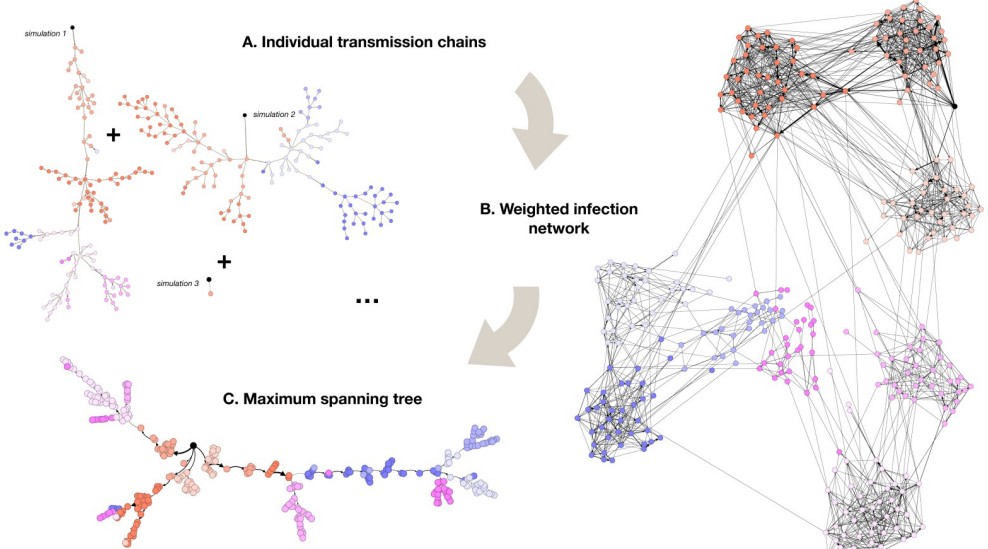

**Fig 2. Construction of infection pathways.** (A) Examples of simulated transmission chains between individuals are shown for a given seed. (B) An infection network $\mathcal{G}_{inf}(s, ct_x)$ built from 135 realisations of the model initialised with the same seed is shown. (C) Maximum spanning tree $\mathcal{T}_{inf}(s, ct_x)$ extracted from the infection network of (B). All results are obtained with $ct_x$ = "Friendship 4d" contacts. Darker edges in (A) correspond to transmission events between different classes. Edge widths in (B) and (C) are proportional to their probability of occurrence $p_\ell(s, ct_x)$. Edges with probability of occurrence <0.01 are omitted for readability in (B). Nodes of the same class share the same color, and the seed is highlighted in black. Visualisations generated with Gephi [51].

The infection pathways defined in this way (networks and trees) describe the progression of the outbreak between students, for each given seed $s$. To adopt a more coarse-grained view, we also build pathways using aggregated transmission chains between classes, in order to represent the pathways of importation of cases into previously unexposed classes. We refer to the S1 Text for additional details on the construction of the four types of infection pathways introduced.

In order to investigate the impact of the type of synthetic data used in the simulations, we compare the infection networks obtained with the same given seed $s$ for two different contact sequences (denoted by $ct_a$ and $ct_b$) using the global cosine similarity defined by:

$$GCS(s, ct_a, ct_b) \equiv GCS(\mathcal{G}_{inf}(s, ct_a), \mathcal{G}_{inf}(s, ct_b)) = \frac{\sum_{\ell \in \mathcal{E}_a \cap \mathcal{E}_b} p_\ell(s, ct_a) p_\ell(s, ct_b)}{\sqrt{\sum_{\ell \in \mathcal{E}_a} p_\ell(s, ct_a)^2} \sqrt{\sum_{\ell \in \mathcal{E}_b} p_\ell(s, ct_b)^2}} \quad (2)$$

where $\mathcal{E}_a$ (resp. $\mathcal{E}_b$) denotes the set of edges of $\mathcal{G}_{inf}(s, ct_a)$ (resp. $\mathcal{G}_{inf}(s, ct_b)$). Note that we can also compute the global cosine similarity between pairs of infection trees. This global cosine similarity takes values between 0 and 1 and provides a measure of overlap between two infection networks (or trees). It is equal to 0 if and only if the networks (or trees) compared have no edges in common, and reaches 1 if and only if the two networks or trees are identical in their edges and respective weights.

## Results

As already described in [7], the total daily interaction time within each class is larger than between pairs of classes (see Fig 3A for day 2). More interaction time is also observed between classes of the same specialty than between classes of different disciplines. This block diagonal structure is correctly reproduced by the synthetic contact networks (Fig 3B and 3C). The global school activity timeline (total interaction time per time window of 15 minutes) is also well reproduced by both types of synthetic contact data (Fig 3D).

As discussed above, the parameters of the friendship-based approach are tuned to reproduce the observed distribution of local cosine similarities in the contact links of each student in different days. We obtain as optimal parameters $f = 0.8$ (average fraction of friendship links included), $1 - p = 0.6$ (probability of drawing remaining links randomly) and $p_{tr} = 0.75$ (probability of adjusting for transitivity). We show in Fig 3E that the resulting distribution of similarities mimics indeed well the empirical one, and Fig 3F shows that the class-mixing-based approach leads instead to a qualitatively different picture with only small values of the similarities between the contact links of each individual in different days. This picture is confirmed by Fig 3G, which displays the global cosine similarity (calculated by applying Eq 2 between the daily contact networks) of consecutive days of the complete contact sequences: the values obtained with contact sequences built using the friendship-based approach are close to the empirical ones, while the class-mixing-based approach leads to very small values. We refer to the S1 Text for figures showing how the two approaches reproduce other properties of the empirical contact networks. For instance, friendship-based contact sequences closely preserve the fraction of repeated contacts, while class-mixing-based contacts underestimate it significantly (see S1 Text, Fig M). Friendship-based contacts additionally perform better than class-mixing-based contacts in reproducing the number of triangles and transitivity, within class and between classes (see S1 Text, Fig N). They also reproduce more closely than class-mixing-based contacts the degree distribution within class (number of contact links within the class of each individual, see S1 Text, Fig O) observed in empirical contacts. Finally, class-mixing-based and friendship-based contact sequences preserve equally well the distributions of node

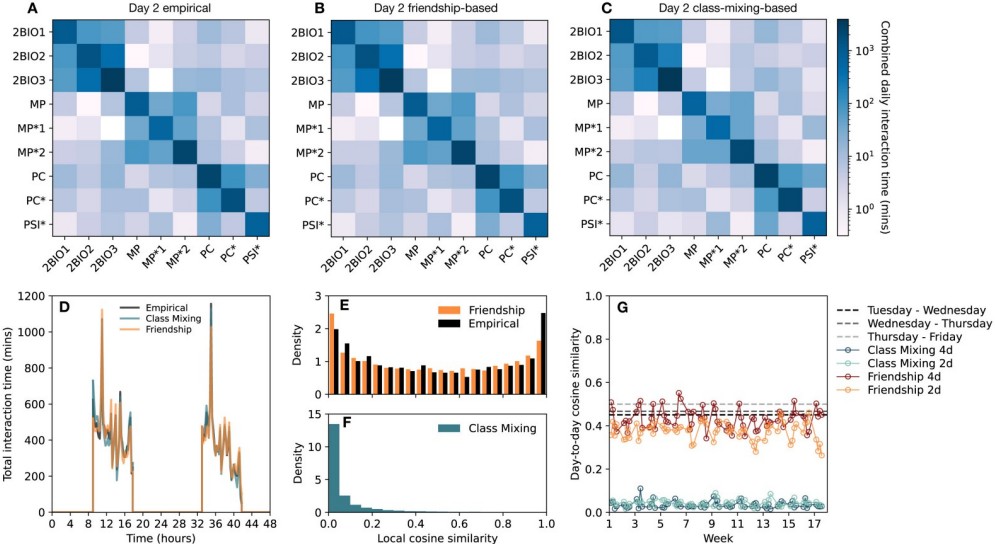

**Fig 3. Comparison between the different contact sequences.** (A) Daily total time measured in contact within and between classes for the recorded contacts on day 2. (B) Same as (A) for the corresponding friendship-based contacts. (C) Same as (A) for the corresponding class-mixing-based contacts. (D) Total time measured in contact between all individuals in the school on successive 15 minutes time steps on days 2 and 3 for the three types of contacts (empirical and two types of synthetic data). (E) Distribution of students' local cosine similarities for each pair of days observed in the empirical contacts (black), together with the same distribution obtained with the friendship-based algorithm with optimised parameters, averaged over 10 realisations. (F) Same as (E) obtained instead from 10 realisations of the class-mixing-based approach. (G) Global similarities between the daily contact networks of consecutive days (computed by applying Eq 2 to the contact networks), for contact sequences obtained with different versions of the algorithm (each color corresponds to one single iteration of the contact sequence).

strengths (see S1 Text Fig O), link weights and number of distinct contact events per pair and day (see S1 Text Fig P), as well as characteristic features of the contacts such as the total daily time spent interacting, number of nodes and number of links (see S1 Text Fig N).

Let us now discuss and compare the outcomes of the epidemic spreading simulations performed on the various contact sequences of Table 1. Fig 4 first focuses on the resulting distributions of epidemic sizes, by showing them as violin plots in Fig 4A, 4B and 4C. The

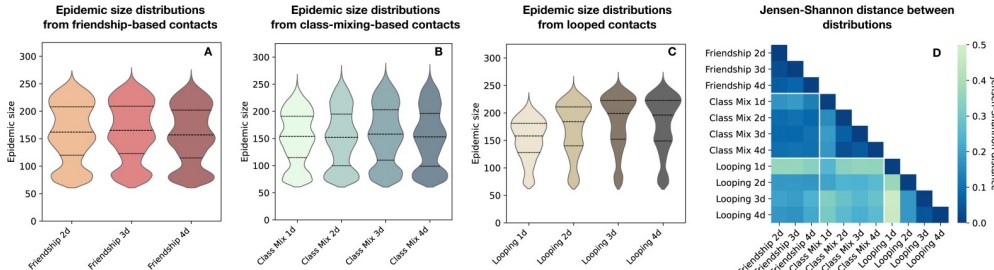

**Fig 4. Epidemic size distributions.** (A) Distributions (Gaussian kernel density estimations) of the final epidemic sizes obtained with friendship-based contact sequences. (B) Same as (A) for class-mixing-based contact sequences. (C) Same as (A) for looped contact sequences. The distributions are computed over simulations leading to a fraction of infected individuals larger than 20% (over 120 days) in order to better highlight differences between the distributions. Results including all simulations are shown in the S1 Text, Fig S and T. The first and third quartiles (25% and 75%) are indicated with dotted lines while the median is shown with a dashed line. (D) Jensen-Shannon distance between all pairs of distributions. For each contact sequence, 150 simulations are conducted for each of the 325 students as seed (48, 750 simulations for each contact sequence).

distributions obtained with the three friendship-based contact sequences (Fig 4A) are visually very similar. This is confirmed quantitatively by the low Jensen-Shannon distances [49, 50] between these three distributions (Fig 4D). Looped contacts instead lead to epidemic size distributions with a different shape (Fig 4C), as confirmed from the Jensen-Shannon distances (Fig 4D). The distance with the epidemic size distributions obtained with the friendship-based contact sequences is particularly large when only one day is looped over. As the comparison of Fig 4A and 4C indicates, this is due to an overall shift of the distribution to lower values. This shift is due to the restricted number of propagation pathways when the same contact patterns are repeated every day ("Looped 1d" case). When two days or more are included in the looped data, the range of epidemic size values reached by the simulations becomes similar to the one obtained with the friendship-based data. However, the shape of the distribution remains different: large outbreaks are in fact more probable, and outbreaks of small and intermediate sizes are much less probable. Finally, friendship-based and class-mixing-based contact sequences lead to visually rather similar distributions of epidemic sizes, with close values of the medians. Differences however emerge for large epidemic sizes, which are more probable when using contact sequences that preserve the balance between friendships and casual contacts.

To go beyond these distributions, we also characterise and summarise the propagation paths in the population, for each seed $s$ and contact sequence $ct_x$ considered, by the infection networks $\mathcal{G}_{inf}(s, ct_x)$ and trees $\mathcal{T}_{inf}(s, ct_x)$ (computed between students on the one hand and between classes on the other hand), as described in the Methods section. We present here the analysis of the infection networks (Fig 5) and refer to the S1 Text for the same analysis concerning the infection trees.

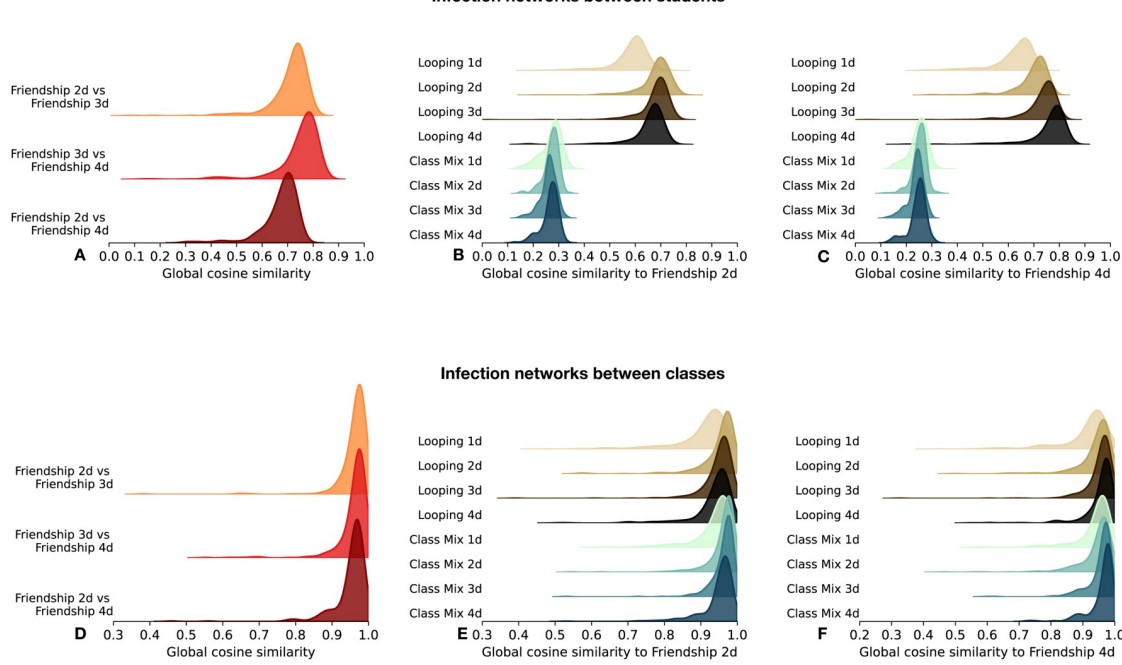

**Fig 5. Pairwise comparisons of simulated infection networks between students and classes.** (A) Distributions over all seeds of the global cosine similarities $GCS(\mathcal{G}_{inf}(s, ct_a), \mathcal{G}_{inf}(s, ct_b))$ are shown for infection networks obtained from pairs of contact sequences $ct_a$ and $ct_b$ in "Friendship 2d", "Friendship 3d" and "Friendship 4d" for infection networks between students. (B) Distributions of $GCS(\mathcal{G}_{inf}(s, ct_a), \mathcal{G}_{inf}(s, ct_b))$ over all seeds $s$ for different sequences $ct_b$ (class-mixing-based and looped contacts) with $ct_a$ fixed to "Friendship 2d" for infection networks between students. (C) Same as (B) with $ct_a$ fixed to "Friendship 4d". (D) Same as (A) for infection networks between classes. (E) Same as (B) for infection networks between classes. (F) Same as (C) for infection networks between classes. For each contact sequence, infection networks are obtained from 150 simulations for each seed, and each of the 325 students are successively considered as seed $s$.

Fig 5A shows that the infection networks between students resulting from simulations performed on friendship-based contact sequences generated from two, three or four days of data and with the same seed are highly similar for almost all seeds (Fig 5A shows the distribution of $GCS(s, ct_a, ct_b)$ for all seeds $s$ and different pairs of contact sequences $ct_a, ct_b$): the number of days used when building the contact sequences does not impact strongly the propagation patterns in the population. Contact sequences built from the class-mixing-based approach lead to infection networks having significantly lower similarity with the infection networks of the friendship-based synthetic data (Fig 5B and 5C). Note that the distributions of $GCS(s, ct_a, ct_b)$ when $ct_a$ and $ct_b$ are both class-mixing-based contact sequences, but built using different numbers of days, peak at approximately 0.4 (see S1 Text Fig X): a quite high diversity of infection networks is thus also obtained within the class-mixing-based approach. These results suggest significant differences in the detailed spreading patterns observed with the two types of synthetic data. On the other hand, simulations on looped contact data are again rather similar to the ones of friendship-based contact sequences (Note also that, for $ct_a$ and $ct_b$ both looped contact sequences –built with different numbers of base days– the infection networks are highly similar, with distributions of $GCS(s, ct_a, ct_b)$ peaking between 0.75 and 0.9, see S1 Text Fig Y).

The results concerning the similarities between infection networks might be put in relation with the distributions of similarities of the individual contact links in different days: they are indeed close for looped and friendship-based synthetic data, while they are very different for class-mixing-based data (Fig 3E, 3F and 3G). This is consistent with the fact that the number of days looped over has an additional impact on the similarity between the resulting infection networks and the ones of the friendship-based data. In particular, infection networks obtained on "Looped 1d" synthetic contact sequences are significantly more dissimilar than for the rest of the looped contact sequences, due to the lack of diversity of these contact sequences. These contact sequences indeed inherit only the contact links and properties of a single day of measurements and lack a realistic diversity in contact behaviour (the local cosine similarities between days are all trivially equal to 1). Looping over as many days (or more) than used to produce the friendship-based contact data leads to infection networks more similar to the ones obtained with the friendship-based data (e.g., Fig 5C indicates an increasing similarity with the infection networks of friendship-based contact sequences built from four base days as the number of days looped over increases from one to four).

When considering the seed wise infection networks between classes, we obtain high values of the global cosine similarities (Fig 5D, 5E and 5F) regardless of the types of contact sequences used. These epidemiological outputs are thus not affected by the distinction between friendship and casual links. Only a slight shift to lower similarity values can be observed with the "Looped 1d" contacts. This can be caused by pairs of classes that do not interact at all on the day looped over, causing slight differences in spreading patterns. Interactions between these pairs of classes are instead observed as soon as two days of empirical data are taken into account. Note that class-mixing-based contact sequences generated from a single base day of data are less affected by this issue, because the stochastic character of the algorithm creates many different pathways between the other classes, and overall produces enough pathways between these non-interacting classes through a third class. The presence of outliers in these distributions (as evidenced by the tails of the distribution) reveals that the propagation paths of the disease between classes remain sensitive to the type of contact sequences considered for a minority of the seeds.

We refer to the S1 Text for additional results teasing out the individual impact of the friendship-preserving parameter $f$ and transitivity preserving parameter $p_{tr}$ on epidemic size distributions and infection pathways.

## Discussion

We have proposed and explored two approaches to generate synthetic contact sequences with realistic statistical properties between students in a school. Both preserve the class-mixing matrices (i.e., the amount of interactions within each class and between pairs of classes) observed daily during the data collection, as well as the global timeline of activity driven by the underlying timetable. The class-mixing-based approach creates contact links independently for each simulated day. On the other hand, the friendship-based approach preserves the balance between contacts that reoccur over different days (indicative of friendship links between students) and casual contacts that were observed only on one of the days of the data collection. This balance is preserved both at the global level and in terms of its heterogeneity between students. We have then performed numerical simulations of a model describing the spread of an infectious disease in the population, informed by synthetic contact sequences built using the two approaches and relying on a varying amount of empirical data. Specifically, we have considered the spreading dynamics of SARS-CoV-2, as an example of recent important concern, in particular in school settings. We have compared the outcomes of the simulations at the level both of the distributions of final epidemic sizes and of the most probable infection pathways between students and across classes.

The friendship-based approach successfully reproduces the main features of empirically observed contact patterns in a school population, while the class-mixing-based approach fails to capture the heterogeneity of similarity levels in the contact patterns of individuals on different days. Both types of synthetic contacts accurately represent the global features of the school where the base data was collected (class-mixing matrix, activity timeline). By preserving as well the balance between friendship relations that lead to repeated contacts and casual encounters, synthetic contact sequences generated with the friendship-based approach provide realistic contact data that can be used for modelling purposes. We note that our work notably differs from several other mechanisms that have been proposed to go beyond the simple looping procedure to extend contact network data while preserving some properties of the data, either in a general context [39] or for epidemic modelling purposes [13, 24]. In particular, the approach of [39] does not preserve class-mixing patterns and does not take into account the existence of repeated contacts (friendships), leading to more random interaction patterns than actually observed. On the other hand, references [13] and [24] focused only on reproducing the global fraction of repeated contacts (whatever their durations) between daily networks (in [13], by a reshuffling of nodes that led to an identical underlying structure in successive days, in [24] by a non-optimized ad-hoc procedure), without dealing with the population heterogeneity. Here, by considering the whole distribution of local cosine similarities between the neighborhood of individuals in different days, and by performing an optimization of the parameters of the algorithm, we can create synthetic contact sequences that faithfully reproduce the heterogeneity of contact behaviours in the population. Finally, several mechanisms to generate realistic synthetic contact data have also been devised to correct for incompleteness of data due to diverse types of sampling [52–55]. These methods however are designed to reconstruct missing contacts occurring alongside empirically recorded contacts, covering thus the data collection window. The friendship-based and class-mixing-based approaches that we have presented fill a different purpose, and instead longitudinally extend limited existing data.

When feeding synthetic data to numerical simulations of spreading models, taking into account the presence of friendships, which lead to repeated contacts with correlated durations in successive days, impacted mostly the transmission patterns at the level of students. Indeed, infection pathways between students are consistently dissimilar when feeding the model with either friendship-based or class-mixing-based contact sequences. Preserving the balance

between friendships and casual encounters is therefore essential to accurately characterise disease spreading patterns between students. Previous works have already noted that contact repetition must also be captured when building daily contact networks [33] from survey data [35, 36]. In that context, ignoring the reported repetition of contacts [31] overestimates the weekly number of distinct contacts per individual. This in turn provokes transmission opportunities that would not occur otherwise, resulting in simulated higher attack rates and lower extinction rates [33]. In the present study, which considers the specific context of interactions in a school, we do not observe the effects reported in Ref. [33], although friendship-based and class-mixing-based approaches indeed lead to different distinct numbers of contact per individual over a set period (e.g., one week). This can be explained by the rather high density of contacts among individuals in the school: the repetition of contacts in this context does not hinder the spread of the disease as a sufficiently high number of distinct contacts between students is present in all cases (see S1 Text Fig Q). Finally, it has been discussed that contact repetition must be considered to accurately identify superspreaders and super-spreading events from contact data [34]. While our study does not explicitly investigate this aspect, we expect the friendship-based contact data to faithfully capture super-spreading dynamics thanks to the fact that it preserves the correlations between contact patterns in different days.

Changing scale from the individual to the classes, we find that preserving class-mixing patterns and global activity levels throughout the day is sufficient to predict the spread of the disease at the scale of classes. Consequently, while taking into account friendship links in synthetic contact data is necessary to evaluate localised and fine grained mitigation strategies targeted to students (e.g. reactive testing of contacts), larger scale protocols targeting classes (such as reactive class closure), grades (closure of all classes of a grade) or the entire school (school closure) can be evaluated using synthetic contact data generated with the class-mixing-based approach. This is in line with the negligible impact of randomising contacts per class on average transmissions found in Ref. [9], and our analysis confirms this in the case of class-based infection pathways.

Our study also highlights consequences of looping over the data, an approach commonly used to inform models with high-resolution data. We note two opposite effects at play. Firstly, looping over recorded data limits the pool of susceptible individuals an infectious individual can pass the infection to (see S1 Text Fig Q). This results in a general shift of epidemic size distributions to lower sizes. This effect is strongest when the number of distinct contacts an individual has over their infectious period is low. In the looped contacts considered, the contact networks are sufficiently dense that this effect is insignificant as soon as two days are looped over. A second effect instead causes the inflated outbreaks observed. The periodic recurrence of all contacts can reinforce transmission routes as identical contacts reoccur during the infectious period. This effect counteracts the depletion of susceptible individuals. In our simulations, we find that it dominates as soon as two days of data are looped over, yielding inflated epidemic sizes. Larger epidemic sizes are known to occur when repeated contacts are not included [33], when contact duration are averaged over the contacts [9, 13] and where the network of contacts is discarded in favor of an all-mixing approximation [29]. The larger outbreaks observed with the looped contacts are instead attributed here to the reinforcement of contacts looped over, in a context where sufficient contacts allow unhindered propagation.

Finally, our results contribute to the design of efficient data collection windows. Both infection pathways and epidemic size distributions obtained from friendship-based contacts remain highly similar to each other as two, three or four base days are used to generate the synthetic contact sequences. Additionally, infection pathways built from friendship-based contacts obtained with two days of data remain similar to pathways built from looping over two, three or four days of data. Friendship-based contact sequences useful for modelling purposes may

therefore be obtained from two days of data. This is in line with previous results showing that two days of recordings are sufficient to accurately predict the epidemic threshold in a temporal network of contacts measured over three days in a workplace context [56]. We note that, as contacts in schools or workplaces are dictated by timetables and shifts, one day of data may already provide representative global features of the contacts (classes, or groups interacting more tightly, structure of a typical day). A minimum of a second day is then necessary to distinguish stable relationships from casual contacts.

Our work presents several limitations worth discussing. First, the friendship-based approach associates to each synthetic contact link a total daily duration and timeline observed in the class, or pair of classes of the contact. This pool of weights and timelines may be too limited when few interactions occur, leading to an unrealistic lack of diversity in the timelines. This does not impact spreading dynamics for diseases progressing slower than the timeline time step [13], as is the case here. In the case of faster disease spread, it may become necessary to generate synthetic timelines from observed timestamps [52] in order to avoid unrealistic repetitions. Second, the numbers of contacts are exactly preserved for each class and pair of classes. Pairs of classes not interacting at all during the data collection period are thus also not interacting in the synthetic data, which may not be realistic over longer periods. In our specific case, this does not affect the friendship-based approach as all classes have interacted by the end of the second day. In other settings, care should be taken to ensure enough contacts are observed to populate the class-mixing matrix. Moreover, the number of days a contact must reoccur over to be considered a friendship was fixed to two, regardless of the time spent interacting on either days. Further tuning of this criteria may be necessary, e.g. for a longer data set in which even casual contacts might be repeated. Finally, further validation of our algorithm is limited by the absence of longer recordings gathered in the school setting. Acquiring such data sets would be highly valuable for future research.

We finally note that the friendship and class-mixing-based approaches extend existing short recordings in the school context to inform models with realistic contact inputs. Generalising these to other contexts might require to capture additional contact characteristics. For example, synthetic contacts in the healthcare setting should capture the turnover of patients due to intakes and discharges of patients, as well as staff shifts. Our approach would therefore need to be modified to capture these elements in order to extend contact data measured e.g. in healthcare settings [16].

## Conclusion

The friendship-based and class-mixing-based approaches provide modellers with generalisable methods to generate synthetic contact sequences over long time-scales from existing data. The friendship-based approach captures repeatability in contacts inherent to social behaviour, a feature crucial for the prediction of infection pathways between students. This contributes to a growing toolkit allowing modellers to inform agent-based models with data of increasing realism without the need for further expensive data collection.

## Supporting information

**S1 Text. A separate pdf file is provided containing complementary technical details and analyses.**
(PDF)

## Acknowledgments

We thank Diego Andrès Contreras for helpful discussions.

## Author Contributions

**Conceptualization:** Alain Barrat, Vittoria Colizza.

**Funding acquisition:** Alain Barrat, Vittoria Colizza.

**Investigation:** Lucille Calmon, Elisabetta Colosi, Giulia Bassignana.

**Methodology:** Lucille Calmon, Giulia Bassignana, Alain Barrat, Vittoria Colizza.

**Software:** Lucille Calmon, Elisabetta Colosi, Giulia Bassignana.

**Supervision:** Alain Barrat, Vittoria Colizza.

**Visualization:** Lucille Calmon.

**Writing – original draft:** Lucille Calmon.

**Writing – review & editing:** Lucille Calmon, Elisabetta Colosi, Giulia Bassignana, Alain Barrat, Vittoria Colizza.

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
