## [Decision Letter · Decision Letter 0]

17 Sep 2024

Dear Dr. Colizza,

Thank you very much for submitting your manuscript "Preserving friendships in school contacts: an algorithm to construct synthetic temporal networks for epidemic modelling" for consideration at PLOS Computational Biology. As with all papers reviewed by the journal, your manuscript was reviewed by members of the editorial board and by several independent reviewers. Based on the reviews, we are likely to reconsider this manuscript for publication, providing that you modify the manuscript according to the reviewers' recommendations through a moderate revision.

Sincerely,

Yamir Moreno

Academic Editor

PLOS Computational Biology

Virginia Pitzer

Section Editor

PLOS Computational Biology

Reviewer's Responses to Questions

**Comments to the Authors:**

Reviewer #1: Study explores the use of an algorithm to recreate contact networks over multiple days via a mix of repeated contacts and random casual contacts. Some minor comments

1. Line 31, “clear impact on the infection pathways between students” this phrasing is unclear. The details are elaborated in the paper but abstract should capture specific details of key findings.

2. Figure 1 panel 2, the details of the data set was illustrated but it's hard to interpret the meaning. Why not provide a sample table of values and illustrate the connections between persons e.g. A, B, C. There can be a column to illustrate the corresponding colour and weight of the link.

3. Analysis of COVID-19 transmission and the different vaccine strategies was unnecessary given that the focus of the paper is on recreating contact networks. Instead a simple transmission model would suffice. Adding too much disease characteristiscs would mask the differences between the networks.

4. The infection risk from multiple 20 sec interactions vs a prolong interaction might differ. Would be interesting to see if there's any variation between the networks on this. From the current plots, the measure of similarity is based on cumulative contact link weights. If the risk of infection "resets" at every exposure (i.e. risk of infection now is independent on previous interactions), then this might change the findings.

5. Line 306, why is the risk of infection not necessarily equal? Is it because of age related susceptibility or infectivity factors? Though age is not a factor mentioned in this study. Or is it due to the absence of record captured by another individual?

6. Figure 4c, when the contacts are repeated why does it result in a larger outbreak? The authors sought to explain in Line 505 but intuitively if I have 5 unchanging contacts and the probability of infection is 1, then i will always infect 5 individuals. If I have 5 changing contacts each day and the probability of infection is 0.2 over 5 days, then my average number of infections wlll also be 5 but could be as high as 25.

Intuitively, repeated contacts reduces the outreach of the diseases to other parts of the network and can end prematurely due to clustering. This is a major area to elaborate further.

Reviewer #2: Summary & Overall Impression

In this manuscript, the authors propose two algorithms to extend short-term high-resolution contact data to longer time horizons. The main innovation in these algorithms is the possibility of preserving specific details of the contact patterns, such as class-mixing probabilities and repeated contacts, termed friendships in the manuscript. The authors demonstrate the practical relevance of preserving these features by conducting agent-based simulations of the spread of SARS-CoV-2 in a school setting based on face-to-face contact data that has been synthetically extended using their method.

The manuscript makes a significant methodological contribution to the study of temporal networks and face-to-face contact data, especially as applied to infectious disease modeling. The writing is clear and concise throughout the manuscript. The presented methods and the assessment of the influence on agent-based models are technically sound. In this context I want to highlight that the authors use a more realistic compartmental model, instead of a toy model like SIR, thereby facilitating the interpretation of their results in practically relevant contexts. I also found the comparison of outbreak sizes on the level of distributions, instead of low-dimensional summary statistics very insightful.

While this manuscript strikes me overall as a strong contribution that will be of interest to a variety of readers, I want to draw the authors’ attention to the following points.

Major Comments:

1. The method that the authors use to tune the algorithm’s parameters is based on optimizing the Jensen-Shannon divergence between the distributions of local cosine similarities (eq. 1) of the empirical and synthetic networks. This appears to be an ad hoc choice, and I wonder whether this choice of objective function can be justified theoretically, e.g., in terms of beneficial properties for the optimization process or from the perspective of an underlying implicit probabilistic model. Providing the reader with the underlying reasoning would help them to understand the choice of parameters in applications and also facilitate follow-up research on the proposed algorithm.

2. Compared to the class-mixing-based approach the friendship-based approach makes two main modifications: It aims to maintain friendship ties and it aims to maintain network transitivity. Both network transitivity and the presence of temporal correlations between links influence how a contagion process unfolds on a temporal network. I believe exploring these effects not only in combination but also in isolation, would strengthen the manuscript by clarifying further which aspects of the empirical contact data need to be preserved and which should be randomized to obtain realistic synthetic data. Extending the analysis presented in the manuscript to the cases p_tr=0, f>0 and p_tr>0, f=0 could, for example, be a first step in this direction.

Minor Comments

As most of the box plots in Figure 5, Panels E and F are fully above a cosine similarity of 0.8, it would be easier to see the differences between different synthetic data generation mechanisms, if the y-axis limits were adapted to the range 0.75 – 1.0 or if a zoom into this region would be provided as an inset.

Reviewer #3: The manuscript describes a method to synthesize contact patterns by temporally extending empirical data from schools. Their method accounts for repeated contacts by distinguishing between friendship and casual contacts. They also propose a method that only preserves the class-mixing patterns. The authors argue that their synthetic contacts preserve multiple features of the empirical dataset, that individual infection pathways differ between their methods but between-class pathways are similar, and that using 2 days of data produces similar outcomes as of using 3 or 4 days.

The manuscript is overall clear and well written, and the methods look scientifically sound. The authors show awareness of the surrounding literature, fitting their work into a relevant gap. Their method can be a great tool for studying local outbreaks in school settings. Worth mentioning, the GitHub repo with the code and data is very organized and easy to navigate. I comment below about a few issues, which I consider to be minor.

- Ideally, it would be great to validate the proposed model with a dataset that spans for a longer time frame – say, as long as a typical outbreak would last for. By this, I mean to check for how long the synthetic contacts still reproduce the features of the dataset, as friendships and casual contacts evolve over time in schools. I understand that this requires a long-term dataset, spanning time frames longer than a week. Is such a dataset available? If so, using this to validate the model would be an important future step, and the authors could mention this in discussion. Otherwise, adding this limitation to the paper's discussion could be a push for the collection of longer-term contact patterns in schools. I understand that this issue is beyond scope and does not impair the worthiness of the manuscript.

- Figure 4.D). While I understood the main message from panels A, B and C, panel D was more difficult for me to read. Checking the bubbles and their sizes was rather distracting, so can the authors consider an alternative visualization? Since the idea is to compare different distributions in pairs, maybe a triangular matrix would be more effective (this is, each row and column represents one approach, and the matrix entry, which could be color-coded, represents the Jensen-Shannon distance). Another possibility would be to use bars instead of the bubbles.

- Figure 5. While all panels in Fig. 5 represent the same quantity (cosine similarity between infection networks), the multiple plot styles employed made it difficult for me to decipher the message. I believe that using a single plot style would be better, so the reader does not have to learn how to read the data in different ways, making the main message clearer. I particularly prefer the filled curve style of panels B and C.

- On line 108: I guess the word “distributions” is repeated.

- On line 286: “when a susceptible individual is in contact with an infection one” – perhaps you meant “infectious”.

**Have the authors made all data and (if applicable) computational code underlying the findings in their manuscript fully available?**

Reviewer #1: Yes

Reviewer #2: Yes

Reviewer #3: Yes

PLOS authors have the option to publish the peer review history of their article (what does this mean?). If published, this will include your full peer review and any attached files.

Reviewer #1: **Yes: **Rachael Pung

Reviewer #2: No

Reviewer #3: No

Figure Files:

Data Requirements:

Reproducibility:

References:

---

## [Decision Letter · Decision Letter 1]

20 Nov 2024

Dear Dr. Colizza,

We are pleased to inform you that your manuscript 'Preserving friendships in school contacts: an algorithm to construct synthetic temporal networks for epidemic modelling' has been provisionally accepted for publication in PLOS Computational Biology.

Best regards,

Yamir Moreno

Academic Editor

PLOS Computational Biology

Virginia Pitzer

Section Editor

PLOS Computational Biology

Feilim Mac Gabhann

Editor-in-Chief

PLOS Computational Biology

Jason Papin

Editor-in-Chief

PLOS Computational Biology

Reviewer's Responses to Questions

**Comments to the Authors:**

Reviewer #2: The authors have addressed all my comments. I thank the authors especially for their thorough investigation of my comments regarding the distance measure and disentangling the role of clustering and friendship ties. I have read the results with great interest.

Reviewer #3: The authors have addressed all of my comments. The new figures are considerably more readable, and the information is now clearer in this version. I congratulate the authors for the work.

**Have the authors made all data and (if applicable) computational code underlying the findings in their manuscript fully available?**

Reviewer #2: Yes

Reviewer #3: Yes

PLOS authors have the option to publish the peer review history of their article (what does this mean?). If published, this will include your full peer review and any attached files.

Reviewer #2: No

Reviewer #3: No

---

## [Editor Report · Acceptance letter]

30 Nov 2024

PCOMPBIOL-D-24-01326R1 

Preserving friendships in school contacts: an algorithm to construct synthetic temporal networks for epidemic modelling

Dear Dr Colizza,

I am pleased to inform you that your manuscript has been formally accepted for publication in PLOS Computational Biology. Your manuscript is now with our production department and you will be notified of the publication date in due course.

With kind regards,

Zsofia Freund
